# Electron Holography for Advanced Characterization of Permanent Magnets: Demagnetization Field Mapping and Enhanced Precision in Phase Analysis

**DOI:** 10.3390/nano14242046

**Published:** 2024-12-20

**Authors:** Sujin Lee

**Affiliations:** Nano Materials Research Division, Korea Institute of Materials Science, Changwon 51508, Republic of Korea; sujin4524@kims.re.kr

**Keywords:** electron holography, permanent magnet, coercivity, noise reduction, demagnetization field, Nd-Fe-B magnet, wavelet hidden Markov model

## Abstract

This review explores a method of visualizing a demagnetization field (*H_d_*) within a thin-foiled Nd_2_Fe_14_B specimen using electron holography observation. Mapping the *H_d_* is critical in electron holography as it provides the only information on magnetic flux density. The *H_d_* map within a Nd_2_Fe_14_B thin foil, derived from this method, showed good agreement with the micromagnetic simulation result, providing valuable insights related to coercivity. Furthermore, this review examines the application of the wavelet hidden Markov model (WHMM) for noise suppression in thin-foiled Nd_2_Fe_14_B crystals. The results show significant suppression of artificial phase jumps in the reconstructed phase images due to the poor visibility of electron holograms under the narrowest fringe spacing required for spatial resolution in electron holography. These techniques substantially enhance the precision of phase analysis and are applicable to a wide range of magnetic materials, enabling more accurate magnetic characterization.

## 1. Introduction

Electron holography [1,2] is a powerful electron-interference technique that can be applied through transmission electron microscopy (TEM). Electron holography overcomes the problem in TEM imaging in which the phase shift information of high-energy electron waves passing through a specimen is lost [3] by enabling access to both the phase and amplitude of the electron wave after it has traversed through the sample. Gabor [4] originally proposed the electron holography technique as a means of overcoming the spherical aberration of the TEM objective lens, which had until recently been a resolution-limiting issue in TEM [5]. Off-axis electron holography is the most widely utilized technique using a Möllenstedt–Düker biprim [6]. The specimen is examined using coherent illumination from an electron source, with the region of interest positioned so that it covers approximately half of the field of view. The electrostatic biprism serves to overlap the object wave passing through the region of interest on the specimen with the reference wave passing through the vacuum. The interference pattern (i.e., electron hologram) must then be processed to retrieve or reconstruct a complex electronic wavefunction that conveys the desired phase and amplitude information about the sample. This operation enables the reconstruction of the phase image, and the detailed methodology for phase retrieval is outlined in Section 2. Regarding magnetic materials, holograms are typically recorded with the objective lens of a conventional microscope turned off, as the strong magnetic field deriving from the objective lens can make the specimen undesirably magnetized in the electron beam direction. To address this issue, a specimen may be placed in a field-free environment using a Lorentz lens (e.g., a mini-lens located under the lower objective pole-piece) to record holograms at high magnification. The practical use of electron holography was achieved through the development of a stable coherent field emission gun (FEG) for TEM [7]. Thereafter, the progress of peripheral techniques included the rapid growth in computer speed and memory [1,8]; the multiple biprism [9,10,11], enabling holograms to be collected free from undesired Fresnel fringes; the sophisticated aberration-correction systems, rendering the reconstructed phase image in improved lateral resolution; the advent of the slow-scan charge-coupled-device (CCD) camera [12] that can record a hologram with sufficient contrast, minimizing specimen/beam drift within a reasonable exposure time; and the recent development of the direct detection camera, which has an excellent modulation transfer function [13].

The reconstructed phase image provides electrostatic potential and/or magnetic induction mapping, as the phase shift in the object wave is induced by the electromagnetic field of the specimen. Electron holography, due to this unique capability, has been extensively utilized for visualizing electromagnetic fields in semiconductor p-n junctions [14,15], examining magnetic domains for hard and soft magnetic materials [16,17,18,19,20], investigating the structural arrangement of magnetic nanoparticles [21,22,23,24,25,26,27,28], and studying magnetic skyrmions and chiral magnetic configurations [29,30,31,32,33,34]. Regarding magnetic materials, a significant focus is the visualization of magnetic domains in permanent magnets, such as Nd_2_Fe_14_B crystals. Known for their high coercivity ($\mu_{0}H_{c}$, where $\mu_{0}$ represents vacuum permeability) and remanence, Nd-Fe-B magnets have been intensively used in traction motors. These magnets derive their magnetic properties from their Nd_2_Fe_14_B phase, characterized by high magnetocrystalline anisotropy (*K*_u_ ≈ 4.5 MJ m^−3^) and saturation magnetization ($\mu_{0}M$ ≈ 1.6 T) [35,36,37]. However, $\mu_{0}H_{c}$ in commercial Nd-Fe-B magnets decreases notably at operational temperatures (~473 K), even after optimal heat treatment. For example, the coercivity of a sintered magnet is approximately 0.2 T at the operating temperature of traction motors (~473 K). $\mu_{0}H_{c}$ can be enhanced by substituting Dy for Nd, resulting in a coercivity of up to 3 T at room temperature due to the improved magnetocrystalline anisotropy. However, this substitution decreases the saturation magnetization of the magnet due to the antiferromagnetic spin coupling between Dy and Fe [35,36,37,38]. Furthermore, from an industrial perspective, Dy is classified as a critical element with limited natural availability, raising concerns about its sustainable use [39,40].

In principle, coercivity represents the critical magnetic field required to induce undesired magnetization reversal. To allow magnetic domains to be reversed under external magnetic fields or thermal fluctuations, the energy barrier associated with *K*_u_ must be overcome. However, magnetization reversal is a complex phenomenon, as it depends not only on the crystallographic microstructure but also on magnetic domain structures, which are sensitive to magnetic dipolar interactions and exchange interactions between neighboring domains. This complexity poses significant challenges for analyzing the coercivity mechanism [41]. Coercivity is determined by the path of least resistance in these mechanisms, in which magnetization reversal can occur either continuously through a coherent or incoherent rotation or discontinuously through a dynamic domain motion. Achieving high coercivity requires impeding magnetization rotation through strong magnetic anisotropy and inhibiting the nucleation or growth of reverse magnetic domains [42]. In particular, the $\mu_{0}H_{c}$ of Nd-Fe-B sintered magnets is predominantly governed by the nucleation mechanism. Hence, reverse magnetic domains tend to nucleate preferentially in the regions of locally weak magnetocrystalline anisotropy, such as the surface of the Nd_2_Fe_14_B grains [43,44]. Therefore, electron holography enables high-resolution analysis of magnetic domain structures by providing detailed information about the magnetization distribution and the direction and strength of the magnetic flux density (*B*) at the nanometer scale, thereby contributing to a deeper understanding of the coercivity mechanism. In addition to electron holography, Lorentz microscopy is another powerful TEM-based technique for imaging magnetic domain structures. By exploiting the lateral deflection of incident electrons due to the Lorentz force, Lorentz microscopy (in Fresnel mode) visualizes magnetic domain walls as black and white contrast lines resulting from the electron deficiency and excess in defocused conditions [45,46]. This capability allows for in situ observation of domain wall motion during magnetization reversal. However, Lorentz microscopy faces challenges in providing quantitative information on magnetic fields and high-contrast imaging of the domain wall under complex magnetization distributions, particularly those with varying magnetic flux directions across domain walls. Other techniques, including micro-SQUID magnetometry [47], magnetic force microscopy (MFM) [48,49], and photoemission electron microscopy (PEEM) [50], are also employed to investigate local magnetic properties. However, these methods often lack the spatial resolution needed to observe magnetic domain structures in nanometer-sized materials or resolve intricate features, such as vortex–core structures.

One of the critical aspects of the coercivity mechanism is the distribution of *H*_d_ within Nd-Fe-B systems, as *H*_d_ contributes to undesired magnetization reversal [51,52,53,54]. While the magnitude and distribution of *H*_d_ in sintered magnets are influenced by factors such as the shape, size, and orientational dispersion of the Nd_2_Fe_14_B crystal grains, an increase in *H*_d_ facilitates the nucleation of reverse magnetic domains, ultimately degrading coercivity [53,54]. The influence of *H*_d_ on magnetization reversal has been investigated through micromagnetic simulations based on Landau–Lifshitz–Gilbert calculations [55,56], primarily focusing on model specimens with simplified geometries [54,57,58,59,60,61,62]. Li et al. [54] showed that the edges and corners of crystal grains, modeled as polyhedral shapes, serve as potential nucleation sites for magnetization reversal. This phenomenon is closely associated with the distribution of the demagnetization field, which is particularly pronounced on the *c*-plane surfaces of the grains. Bence et al. [58] investigated the demagnetization field by calculating its distribution in artificial crystal grains with diameters ranging from 55 nm to 8.3 μm, focusing on the influence of surface grains on magnetization reversal. Despite these simulation studies, experimental methods capable of allowing the direct analysis of *H*_d_ in real magnets remain insufficient. Understanding the distribution of *H*_d_ within magnets provides a crucial indicator for effectively enhancing their coercivity. For example, as mentioned earlier, reverse magnetic domains preferentially nucleate in regions with locally reduced magnetocrystalline anisotropy, such as the surfaces of Nd_2_Fe_14_B grains. To ameliorate this problem, substituting Nd sites in the Nd_2_Fe_14_B lattice with heavy rare earth (HRE) elements, such as Dy and Tb, only on the surfaces of the crystal grains can locally increase the magnetic anisotropy field at nucleation sites by forming an HRE-rich shell with a high-anisotropy field [43,44,63]; this method is known as the heavy rare earth grain boundary diffusion process (HRE-GBDP) [64]. This effect induced by the magnetically “hard” HRE-rich shell can maximize when we know where the *H_d_* is strongly distributed within the magnet. Following the discussion by Li et al. [54], introducing the HRE-rich shell only at the c-plane surface of the grain, showing the strong *H_d_* distribution, required a significant external field for the nucleation of reverse magnetic domains. Hence, the discussion about *H_d_* is important for developing high-coercivity Nd-Fe-B magnets.

Discussion about *H_d_* is also essential for electron holography studies as well as for the permanent magnet industry. In principle, electron holography detects only *B*, which arises from the combined components of magnetization (*M*) and the magnetic field (*H*). Furthermore, there are two sources contributing to *H*: the $H_{d}$ within the specimen and the external stray magnetic field ($H_{s}$) outside it. This makes extracting phase data related to $H_{d}$ a persistent challenge. This problem is common in other electron microscopy and spectroscopy methods. For example, X-ray magnetic dichroism (XMCD) [65,66], spin-polarized scanning tunneling microscopy (SP-STM) [67], and micro-SQUID magnetometry [68] only constitute sensitive magnetization (*M*), which has a complementary functionality to that of electron holography. Magnetic force microcopy [69] does not straightforwardly provide information on the near-surface distribution of magnetization as it measures the magnetic force between the magnetic moment of the tip and the stray magnetic field (*H_s_*) from the specimen. Among the methods that can be used with transmission electron microscopy (TEM) are counted Lorentz microscopy and differential phase contrast (DPC) microscopy, which are useful methods for revealing the magnetic domain structures [70,71,72], but they suffer from the same issue regarding electron holography, i.e., they are only sensitive to *B*, which indicates the difficulty of extracting phase information due to *H_d_* [73]. This importance for both the research fields of microscopy and permanent magnets triggers the study of demagnetization fields determined from electron holography observation (i.e., reconstructed phase images).

In addition, high-precision phase analysis is essential for the accurate mapping of demagnetization fields and the magnetic characterization of permanent magnets. The phase detection limit is determined by factors including the electron count per resolved pixel, the camera’s detection quantum efficiency, and the fringe contrast of the electron (i.e., visibility, *V_obs_*) [8,74]. Applying noise reduction techniques through image processing can greatly enhance the accuracy of phase analysis without altering optical or interferometry parameters. In principle, the use of a long exposure time to achieve higher electron doses may be the easiest approach to improving the phase detection limit. However, this condition can lead to unwanted events, such as surface contamination and specimen drift. Therefore, imaging processing, including sparse coding [75,76,77] and tensor decomposition [78], have proven to be effective for low-dose holograms and in situ experiments. Tensor decomposition reduces noise by separating data into low-rank components and residual noise. The low-rank components capture the dominant structural or physical features of the data, while high-frequency noise is isolated in sparse or less significant components. Noise reduction can be effectively performed using tensor decomposition, thereby preserving the essential information [79,80,81]. As a result, Nomura et al. [78] reported that the employment of tensor decomposition allowed the successful extraction of reasonable phase information from a low-dose electron hologram of the p-n junction. Also, sparse coding enhances electron holograms by representing the data through a sparse combination of significant patterns (learned dictionary elements) [82]. This method emphasizes essential signals while reducing irrelevant or noisy components [82,83,84]. Following the methods outlined in the study by Anada et al. [75,76], the electrostatic potentials in GaAs-based p-n junctions were clearly visualized from low-dose holograms. Takahashi et al. [77] showed that denoising using sparse coding successfully removed the inevitable phase jumps in phase images of Pt nanoparticles that make it much more difficult to quantitatively analyze local phase values, and details about this will be discussed in Section 4.

In addition, Midoh and Nakamae [85,86] developed an advanced noise reduction technique known as the wavelet hidden Markov model (WHMM), which minimizes noise through the statistical control of wavelet coefficients. Unlike conventional thresholding, which can eliminate weak signals along with noise [87], the WHMM employs Markov parameters to distinguish between the signal and noise [56]. By employing a model with two hidden states (*L* and *S*) for each pixel, WHMM utilizes the Baum–Welch algorithm to optimize noise suppression. This method enhances noise reduction by probabilistically determining whether a pixel corresponds to an actual signal or noise, allowing for tailored adjustments at the individual-pixel level. Further details can be found in the original work by Midoh and Nakamae [85]. Tamaoka et al. [88] utilized the WHMM in the analysis of holograms experimentally obtained from a multilayered LaFeO_3_/SrTiO_3_ film. Noise reduction improved the quality of the reconstructed phase image, representing a gap in the electrostatic potential at the non-magnetic LaFeO_3_/SrTiO_3_ interface. Despite most previous studies on noise suppression focusing primarily on non-magnetic materials, these denoising methods incorporating the WHMM hold significant potential for broader application in electron holography, particularly in the observation of magnetic domain structures. Note that Nd_2_Fe_14_B crystals present challenges for phase analysis related to magnetic domains due to the poor contrast images created by heavy Nd element absorption. Therefore, noise reduction can be a promising method of improving magnetic domain analysis for Nd-Fe-B-based magnets.

Furthermore, electron holography requires narrow fringe spacing (on the order of 1 nm) for high spatial resolution. However, high-frequency components, which correspond to narrow fringe spacing, tend to be more attenuated than low-frequency components associated with wider fringe spacing. The WHMM can effectively mitigate this issue by applying noise reduction to the complex image generated by the fast Fourier transform (FFT) of the electron hologram during the phase retrieval process. This approach differs from conventional methods, where noise suppression is typically applied to the holograms themselves, as seen in earlier studies [75,76,77,78,85,88]. Therefore, the aim of this review article is to summarize the method of extracting the phase information of *H_d_* within a single-crystalline Nd_2_Fe_14_B thin foil using the reconstructed phase image and the effectiveness of noise reduction based on applying the WHMM to the complex image of a thin foil comprising Nd_2_Fe_14_B crystals for further enhancing phase analysis [89,90].

## 2. Principle of Electron Holography

Before addressing the method of extracting *H_d_* and the WHMM effect, this section outlines the fundamental principles of electron holography. Figure 1 shows a schematic cross-section of a bar magnet magnetized along the *y*-axis, with incident electrons traveling in the −*z* direction. The paths labeled P₀-P and Q₀-Q indicate the trajectories of the electrons. For simplification, it is assumed that the points P₀, P, Q₀, and Q are sufficiently distant from the bar magnet, where the magnetic flux density is negligible. At point P, located beneath the bar magnet in Figure 1, the change in phase of the electron wave passing through the specimen is described by [1]
(1)$$\phi_{P}=\sigma \int_{P_{0}}^{P}V\left(x,y,z\right)dz-\frac{e}{\hbar }\int_{P_{0}}^{P}A_{z}\left(x,y,z\right)dz,$$
where *σ*, *e*, and ℏ are interaction constants that depend on the acceleration voltage applied to the incident electrons, the elementary charge, and Planck’s constant divided by 2π, respectively. $V\left(x,y,z\right)$ represents the electrostatic scalar potential. If the electrical charging, caused by electron exposure, of the specimen is negligible, then this term approximates the mean inner potential of the specimen (*V*_0_) [1]. $A_{z}\left(x,y,z\right)$ is the *z* component of the vector potential (***A***) with respect to the magnetic flux density ***B***. It is important to note that the phase shift arising from *V*$\left(x,y,z\right)$ can be isolated from that due to *A_z_*$\left(x,y,z\right)$ by applying a time-reversal operation with electron waves [91]. In TEM observations, this can be achieved by inverting the specimen’s orientation relative to the incident electrons. Consequently, the phase shift caused by the magnetic field, corresponding to the second term in Equation (1), between two points P and Q below the specimen in Figure 1 is expressed as
(2)$$\phi_{PQ}=-\frac{e}{\hbar }\int_{Q_{0}}^{Q}A_{z}\left(x,y,z\right)dz+\frac{e}{\hbar }\int_{P_{0}}^{P}A_{z}\left(x,y,z\right)dz.$$

Using Stokes’ theorem and the relationship ***B*** = rot***A***, the phase shift $\phi_{PQ}$ can be rewritten as a surface integral involving the *y* component of the magnetic flux density, *B_y_*$\left(x,y,z\right)$:(3)$$\phi_{PQ}=-\frac{e}{\hbar }\iint_{P_{0}Q_{0}QP}B_{y}\left(x,y,z\right)dxdz.$$

Here, the surface integral is evaluated over the closed path $P_{0}Q_{0}QP$.

The electron wave’s phase shift, after passing through the specimen, is recorded in an electron hologram, consisting of interference fringes created with a reference wave unaffected by the electromagnetic field of the specimen [1]. To retrieve phase information, the hologram is digitized. Figure 2 outlines the process of phase retrieval via FFT, illustrating the real (r) and imaginary (i) components of the reconstructed images produced by the inverse Fourier transform (FFT^−1^). Performing an FFT on the hologram generates digital diffractograms (Figure 2b), where a frequency-selection mask isolates a sideband with phase information and shifts it to the center. FFT^−1^ then reconstructs a complex image, separating it into real and imaginary parts (Figure 2c,d). WHMM-based noise reduction, discussed in Section 4, is applied to the real and imaginary parts instead of the original hologram (Figure 2a), resulting in denoised components (Figure 2e,f). The phase image containing the information on the electromagnetic field is then reconstructed using $\tan^{-1}i/r$, as presented in Figure 2g. For additional information on phase retrieval from holograms, refer to Refs. [1,92].

## 3. Mapping of Demagnetization Field Within a Nd_2_Fe_14_B Thin Foil

As elucidated in the introduction, electron holography allows for the determination of phase shifts induced by *B* [1,2]. However, isolating and analyzing *H_d_*, which represents only one component of *B*, remain challenging. This technical constraint (i.e., the inability to directly observe *H_d_*) is also a problem for other approaches, such as XMCD [65,66], SP-STM [67], micro-SQUID magnetometry [68], magnetic force microscopy [69], Lorentz microscopy, and differential phase contrast microscopy [70,71,72]. Moreover, understanding *H_d_* is crucial for developing high coercivity in permanent magnets. The *H_d_* significantly influences the coercivity mechanism, as regions with a high distribution of *H_d_* within the magnet can initiate the nucleation of reverse magnetic domains, resulting in a reduction in coercivity [53,54]. The role of *H_d_* in the coercivity mechanism has been explored through micromagnetic simulations, primarily focusing on simple model specimens [54,57,58,59,60,61,62]. However, investigations of *H_d_* in real magnetic specimens remain limited due to the technical constraints of experimental tools.

This section introduces a method for mapping *H_d_* using phase images, providing a potential solution to address challenges associated with *H_d_* in both the field of electron microscopy and magnetic material engineering.

### 3.1. Method for Determining the Phase Map Concerning H_d_

*B* is a combination of *M* and *H*, which include both the external *H*_s_ and internal *H_d_*:(4)$$B=\mu_{0}M+\mu_{0}H=\mu_{0}M+\mu_{0}H_{s}+\mu_{0}H_{d}.$$

According to Equation (4), the phase shift $\Delta \phi_{B}$ due to *B*, as detected via electron holography, can be divided into phase contributions from magnetization, the stray magnetic field, and the demagnetization field:(5)$${\Delta \phi }_{B}={\Delta \phi }_{M}+{\Delta \phi }_{H_{s}}+{\Delta \phi }_{H_{d}},$$
where ${\Delta \phi }_{M},$ ${\Delta \phi }_{H_{s}}$, and ${\Delta \phi }_{H_{d}}$ correspond to the phase shifts from each source (i.e., *M*, *H*_s_, and *H_d_*). In the Nd₂Fe₁₄B phase, the magnetic parameter $Q\left(=K/K_{d}\right)$ is 4.5, where K is the effective anisotropy constant (taken to be *K*_u_$=4.5\times 10^{6}$ J m^−3^) and $K_{d}$ is the stray field energy coefficient ($K_{d}=\frac{\mu_{0}}{2}{M_{s}}^{2}=1.0\times 10^{6}$) [93,94,95]. A Q value greater than 1 indicates that magnetocrystalline anisotropy dominates the magnetic domain structure [96], suggesting that grains are primarily magnetized along the *c*-axis (the easy-magnetization axis) of the Nd_2_Fe_14_B crystals. The c-axis direction and specimen thickness can be determined via TEM observations.

In reference to Figure 1, magnetization *M* is restricted to Area 2 (i.e., the middle region within $P_{0}Q_{0}QP$), allowing the phase shift ${\Delta \phi }_{M}$ to be determined using a surface integral [90]:(6)$${\Delta \phi }_{M}=-\frac{e}{\hbar }\iint_{area2}\mu_{0}M^{y}dxdz,$$
where $M^{y}$ is the *y* component of the magnetization. This integral covers the entire cross-sectional area of the specimen. The resulting phase shift ${\Delta \phi }_{M}$ is one component of the observed ${\Delta \phi }_{B}$ along the P-Q line in Figure 1. Subtracting ${\Delta \phi }_{M}$ from ${\Delta \phi }_{B}$ yields another phase map representing ${\Delta \phi }_{H_{s}}+{\Delta \phi }_{H_{d}}$, which is the contribution of the stray and demagnetization fields: ${\Delta \phi }_{H_{s}}+{\Delta \phi }_{H_{d}}={{\Delta \phi }_{B}-\Delta \phi }_{M}$. In Nd-Fe-B magnets, this residual phase map offers valuable insights into the demagnetization field within the specimen. To study *H_d_* more comprehensively, the contribution of *H*_s_ must be reduced. To achieve this, the external stray field *H_s_* must be calculated in three dimensions, taking into account the shape of the specimen, thickness, and the c-axis orientation of the Nd₂Fe₁₄B crystal, using the commercial software ELF/MAGIC (ver.2.3.0, ELF Corp., Osaka, Japan). With the three-dimensional distribution of *H_s_*, the phase shift ${\Delta \phi }_{H_{s}}$ can be obtained via surface integrals over the regions outside the specimen (Areas 1 and 3 in $P_{0}Q_{0}QP$ in Figure 1) [90].
(7)$${\Delta \phi }_{H_{s}}=-\frac{e}{\hbar }\iint_{outside}\mu_{0}H_{s}^{y}\left(x,y,z\right)dxdz,$$
where $H_{s}^{y}$ represents the *y* component of the stray magnetic field. It should be noted that the surface integrals for the calculation of the phase shift, such as Equations (6) and (7), yield only the relative phase change between measurement point Q and reference point P. Thus, the value of the phase at point P is assumed to be zero so that $\phi_{PQ}$ simply provides the magnitude of the phase shift between P and Q. To assess the phase-shift images, the offset [i.e., the initial value of the phase determined using Equations (6) and (7)] should be determined for the reference point, such as a point indicated by P. In this case, the offset at the reference points was determined using values derived from an experimentally obtained phase image, with further explanations addressed later.

The phase shift caused by the demagnetization field can be isolated using the surface integrals from Equations (6) and (7):(8)$${\Delta \phi }_{H_{d}}={{\Delta \phi }_{B}-\Delta \phi }_{M}-{\Delta \phi }_{H_{s}}.$$

The effectiveness and validity of this method will be examined in Section 3.2 through its application to a real Nd_2_Fe_14_B thin-foil specimen.

### 3.2. Evaluation of the Method Using a Nd_2_Fe_14_B Specimen

Figure 3a presents a TEM image of the rectangular specimen used for electron holography acquisition. The *x*-*y*-*z* coordinate system in Figure 3 defines the foil plane as being parallel to the *x*-*y* plane, with the electron beam directed along −*z*. TEM imaging and electron diffraction (inset of Figure 3a) identified the *c*-axis orientation as [0.02671, 0.9957, −0.08836] within this coordinate system, indicating that the c-axis aligns closely with the long axis of the specimen, as shown by the yellow arrow in Figure 3a. Figure 3b presents a reconstructed phase image of ${\Delta \phi }_{B}$ acquired from the area shown in Figure 3a via electron holography. The phase shift, illustrated on a color scale, indicates the *y*-axis magnetization in the specimen. The white arrows in Figure 3b denote magnetic flux directions within the specimen (outlined by gray borders) and in the surrounding vacuum. For hologram acquisition, the electron biprism was positioned 1908 nm from the specimen’s right edge (x = 0 nm), with an interference width of 2663 nm. These acquisition parameters, along with the specimen shape, c-axis orientation, and thickness variations (measured via electron energy-loss spectroscopy, the details of which are not shown here), were incorporated into a simulation to determine the three-dimensional distribution of *H_s_* and *M*. As a result, the calculated ${\Delta \phi }_{B}$ in Figure 3c agrees well with the electron holography observation in Figure 3b. This result indicates that reasonable *H_s_* and *M* distribution data can be obtained from this simulation.

Figure 3d shows the phase image representing ${\Delta \phi }_{M}$ calculated using Equation (6). The ${\Delta \phi }_{M}$ was plotted within the region enclosed by black dotted lines in Figure 3, where the artificial phase gap from the surface integral across the specimen’s borders can be disregarded. The phase gradient is positive in the *x* direction due to the magnetization along the *y*-axis. Figure 3e shows the phase image representing ${\Delta \phi }_{H_{s}}$, calculated using Equation (7) for the model specimen. This result was also plotted within the area enclosed by black dotted lines. In the specimen region (the lower part of Figure 3e), the phase gradient is negative along the *x*-axis, reflecting the stray magnetic field opposing the magnetization direction. Figure 3f shows the phase image relevant to ${\Delta \phi }_{H_{d}}$ within the specimen, calculated by subtracting the results of Figure 3d,e from the experimentally determined ${\Delta \phi }_{B}$ of Figure 3b (i.e., by using Equation (8)). The phase gradient in Figure 3f is most significant near the specimen’s edges, while the inner area presents negligible change in the phase. These results indicate the effectiveness of this method, indirectly providing information about the *H_d_* distribution. Note that the offset from the surface integral, representing the initial phase shift at reference point P in Figure 1, was included in the calculations for Figure 3d (inside the specimen) and Figure 3e (outside the specimen), which were determined based on the observations made via electron holography (Figure 3b). Thus, in the plot in Figure 3f that subtracts ${\Delta \phi }_{M}+{\Delta \phi }_{H_{s}}$ from ${\Delta \phi }_{B}$, we assumed no contribution from the offset. However, for a more accurate analysis of ${\Delta \phi }_{H_{d}}$ in irregularly shaped real specimens, the offset requires further consideration, remaining a challenge in phase analysis related to ${\Delta \phi }_{H_{d}}$.

The phase shift ${\Delta \phi }_{H_{d}}$ was converted to the *y*-component of the demagnetization field ($\mu_{0}H_{d}^{y}$, in tesla) using the following relationship [90]:(9)$$\mu_{0}H_{d}^{y}=-\frac{\partial {\Delta \phi }_{H_{d}}}{\partial x}\cdot \frac{\hbar }{e}\cdot \frac{1}{t},$$
where t represents the specimen’s thickness. By applying Equation (9), $\mu_{0}H_{d}^{y}$ was mapped in the thin-foil specimen, as shown in Figure 4a, corresponding to the area enclosed by the blue lines in Figure 3f. At position #1, the $\mu_{0}H_{d}^{y}$ was determined to be −0.38 T, with the negative sign indicating a magnetic field toward the −*y* direction. The $\mu_{0}H_{d}^{y}$ map shows an increase to the right and a decrease to the left of position #1, attributed to the tapered cross-section of the thin-foil specimen fabricated using a focused ion beam. At position #2 (75 nm from #1), the magnitude of $\mu_{0}H_{d}^{y}$ decreased to −0.14 T and further reduced to −0.034 T at position #3 (146 nm from #1). This rate of reduction in $\mu_{0}H_{d}^{y}$ follows the power law of $r_{q_{m}}^{-2}$, where $r_{q_{m}}$ represents the distance from the magnetic charge (magnetic pole), consistent with classical electromagnetism [96]. These findings highlight the utility of this method in extracting the *H_d_* from phase images. Figure 4b presents a result of a micromagnetic simulation (Landau–Lifshitz–Gilbert calculations [55,56]) obtained using EXAMAG LLG code (Fujitsu). The simulation utilized a model specimen approximating the morphological and crystallographic information of the actual specimen, including the specimen’s shape, size, thickness variation, and crystal orientation. The exchange stiffness constant and magnetocrystalline anisotropy for the Nd_2_Fe_14_B phase were set to 12 pJ m^−1^ and 4.5 MJ m^−3^, respectively [94,97]. The simulation result for the $\mu_{0}H_{d}^{y}$ distribution within the specimen shows good agreement with the observation in Figure 4a. Slight deviations in $\mu_{0}H_{d}^{y}$ were observed at positions #1, #2, and #3, with observations lower than the simulation predictions. These discrepancies may arise from uncertainties in foil thickness, crystal orientation, and the geometric characteristics of the specimen. Nonetheless, the obtained agreement within <0.065 T affirms the accuracy of this method for generating a map of *H_d_* using electron holography observations.

## 4. Improving the Precision of Phase Analysis via Reducing the Noise in Electron Holography Observation Using the WHMM

Extracting *H_d_* from phase images involves several analytical steps, including subtraction and differential operation for phase images to derive the $\Delta \phi_{H_{d}}$ and *H_d_*, respectively. These operations may compromise the accuracy of mapping $\Delta \phi_{H_{d}}$. To enhance the quality of the phase image, noise reduction can significantly improve the precision of the method, enabling a more in-depth analysis of *H_d_*. To date, numerous studies have demonstrated improved precision in phase analysis by applying sparse coding [75,76,77], tensor decomposition [78], and the WHMM [85,88,89] to electron holograms. Notably, as explained in Section 1, there is a dilemma regarding the interferometric parameter: while a narrow *s* is essential for achieving high spatial resolution in phase images, it simultaneously leads to a deterioration of phase resolution due to reduced *V_obs_* [1]. To overcome this technical limitation, denoising using the WHMM was applied to the complex images of Nd_2_Fe_14_B thin foil, with variations in the fringe spacing of holograms, distinguishing it from other denoising methods typically used for electron holograms. Therefore, in this Section, we review the effectiveness of applying WHMM denoising to complex images of a thin-foil Nd-Fe-B magnet, highlighting its capability to suppress the artificial phase jumps induced by insufficient *V_obs_*.

For this purpose, the double-biprism system was employed, which allows independent control of the *s* and interference width (*W*), where *W* is the product of fringe spacing and the number of fringes [9]. The upper biprism voltage (*V_BP1_*) varied from −50 V to −150 V in −20 V increments, while the lower biprism voltage (*V_BP2_*) remained fixed at −100 V. This setup resulted in *s* values of 5.2 nm, 3.7 nm, 2.9 nm, 2.4 nm, 2.1 nm, and 1.7 nm, with *W* kept constant at 1344 nm.

### 4.1. Effect of Fringe Spacing on Reconstructed Phase Image

Figure 5a shows a TEM image of the polycrystalline Nd_2_Fe_14_B thin-foil specimen, consisting of multiple grain boundaries and a triple junction. Figure 5b presents the *V_obs_* of electron holograms as a function of the *V_BP1_* (shown on the lower horizontal axis) and *s* (shown on the upper horizontal axis). All holograms were obtained with an electron exposure time (*t_a_*) of 3.0 s. The $V_{obs}$ was calculated using the equation $\frac{I_{max}-I_{min}}{I_{max}+I_{min}}$, where $I_{max}$ and $I_{min}$ represent the maximum and minimum intensities of the electron hologram, respectively [2]. Figure 5b presents the average $V_{obs}$ value determined within the rectangular area enclosed by the yellow lines in Figure 5a. Figure 5c exhibits electron holograms with three conditional *s* values of 5.2 nm, 2.4 nm, and 1.7 nm, demonstrating that $V_{obs}$ decreases as *s* is reduced. According to Chang et al. [98], the $V_{obs}$ is determined by various factors: a time-dependent component associated with instrument instabilities (slower than the exposure time), the spatial coherence envelope of the wave field (including faster instabilities), and the camera’s modulation transfer function (MTF) at the fringe spatial frequency $k_{0}$. When electron holograms are acquired at a constant *V_BP2_* (i.e., with a constant *W*, representing spatial coherency), the observed decrease in $V_{obs}$ can be attributed to the MTF’s frequency dependence [9,13]. Eventually, the $V_{obs}$ diminishes with decreasing fringe spacing, as shown in Figure 5b,c.

The deterioration of $V_{obs}$ impacts the quality of the phase image reconstructed via the FFT process. Figure 6a–c show phase images reconstructed from holograms acquired at fringe spacings of (a) 5.2 nm (*V_BP1_* =−50 V), (b) 2.4 nm (*V_BP1_* =−110 V), and (c) 1.7 nm (*V_BP1_* =−150 V), respectively. In Figure 6, the phase shift is depicted on a color scale, with the specimen border marked by a white dotted line. As previously discussed, the $V_{obs}$ decreased with reducing fringe spacing, as shown in Figure 5b,c. Notably, low fringe contrast in the electron hologram can introduce artificial discontinuities when running the phase-unwrapping algorithm [1], resulting in unintended 2π phase jumps that appear as patches in the colored phase images. As detailed in Section 2, the phase shift is determined by applying $\tan^{-1}i/r$ to the real and imaginary parts of the complex image in Figure 2. The arctangent function inherently produces a “wrapped” phase, constrained within the range of −π to π. A phase-unwrapping algorithm is subsequently applied to extend the phase range, revealing continuous phase changes across the field of view, as shown in Figure 6a. The occurrence of these phase jumps has been elucidated briefly in [1,7]. However, the phase unwrapping was unsuccessful when the $V_{obs}$ was significantly reduced. As shown in Figure 6a–c, the number of artificial phase jumps (patches) increased as $V_{obs}$ decreased. Due to the wedge-shaped cross-section, the thinner right side of the specimen exhibited higher fringe contrast and fewer phase jumps, while the thicker left side showed poorer contrast and more patches.

### 4.2. Evaluation of Denoised Images Using WHMM

To evaluate the effectiveness of the WHMM, the main factor assessed was the reduction in artificial phase jumps. Figure 7a–f present phase images collected under two conditions of *s*: (a–c) 2.1 nm and (d–f) 1.7 nm. The field of view corresponds to the area marked by the blue rectangle in Figure 5a. The “reference measurement” images, shown in Figure 7a,d, were reconstructed from holograms with a long exposure time (*t_a_* = 15 s) and show negligible phase jumps. The “noised” images in Figure 7b,e represent the original phase images without being subjected to WHMM denoising, resulting in many patches during the phase retrieval process due to poor visibility of the interference fringe. After applying WHMM, the “denoised” images in Figure 7c,f show that most patches were eliminated by noise reduction targeting the real and imaginary parts of the complex image (see Figure 2).

Figure 7g,h plot phase shifts measured along the R-S lines in Figure 7a–f, respectively. Black, blue, and red dots represent the phase shifts investigated from the reference-measurement, noised, and denoised images, respectively. Note that the steep phase changes at the specimen borders (x= 170 nm and x= 825 nm) were attributed to the mean inner potential *V*_0_ for the crystal. The phase shift induced by *B* for the Nd_2_Fe_14_B crystal [second term in Equation (1)] is superimposed on the phase shift caused by *V*_0_ [first term in Equation (1)]. However, within the 170 nm < x < 825 nm region, where the specimen is magnetized along the -*y* direction, the phase plot continues to show a downward trend. Regarding the noised images, the blue dots deviate from the reference curve, particularly at *s*
= 1.7 nm due to low $V_{obs}$. The deviation is more pronounced on the left side of the specimen, where $V_{obs}$ decreases in the thicker region. In contrast, the red dots representing denoised images agree well with the black dots representing the reference measurement images in terms of magnitude and smoothness. These results clearly demonstrate the efficacy of the WHMM in noise reduction.

To further assess noise reduction, the peak signal-to-noise ratio (PSNR), defined in Equation (10), was computed for the denoised images [99]:(10)$$PSNR=20\cdot \log_{10}\left(255/\sqrt{1/MN\sum_{i=1}^{M}\sum_{j=1}^{N}{\left(I\left(i,j\right)-G\left(i,j\right)\right)}^{2}}\right),$$
where $I\left(i,j\right)$ represents the intensity at pixel position $\left(i,j\right)$ in either the original (without denoising) or denoised images, and $G\left(i,j\right)$ denotes the intensity of the reference image at the same pixel positions. *M* and *N*, both set to 536, define the pixel dimensions of the image. The *PSNR* was calculated for the region outlined by the yellow lines in Figure 8a, corresponding to the area shown in Figure 7a. Figure 8b plots the *PSNR* as a function of *V_BP1_* (the lower axis) and *s* (the upper axis). Open squares indicate the *PSNR* of the original phase images, while closed squares represent the *PSNR* of the denoised images. For the original images, the *PSNR* decreased as the *s* narrowed due to the decline in $V_{obs}$. At *s* = 1.7 nm, the *PSNR* for open squares reached −1.9, indicating that the signal was significantly weaker than the noise. Following WHMM-based noise reduction, the *PSNR* increased across all fringe spacing conditions, as indicated by the closed squares in Figure 8b. Notably, the *PSNR* at *s*
= 1.7 nm for denoised images became comparable to that of noised images at *s*
= 2.9 nm. Although reduced fringe spacing deteriorates $V_{obs}$ (responsible for the sensitivity of phase detection), noise reduction improves $V_{obs}$. These results show that WHMM-based noise reduction is beneficial for electron holography at high spatial resolutions. To evaluate this technique’s impact on magnetic domain analysis, Figure 8c compares the original phase image (labeled “noise”) at *s*
= 2.4 nm with the denoised image. The field of view in Figure 8c corresponds to an enlargement enclosed by the dashed line in Figure 8a. The original image, representing a single magnetic domain magnetized in the *y* direction, contains phase discontinuities (patches) due to imperfect phase retrieval. Applying noise reduction removed these discontinuities, as exhibited in the right panel labeled ‘denoised’. The residual patch in the denoised image, located in the center-left area, corresponds to a region where the specimen’s thickness was reduced during sample preparation. Thus, WHMM-based noise reduction enables a clearer analysis of magnetic domain structures and magnetic flux density in electron holography.

## 5. Conclusions

Electron holography is an indispensable tool for obtaining information on magnetic flux density and visualizing magnetic domain structures in permanent magnets by determining the phase shift of an electron wave passing through the specimen. This review summarized the studies on Nd_2_Fe_14_B crystals using electron holography, consisting of (1) a method for extracting phase information about demagnetization fields and (2) a method for improving the precision of phase analysis via noise reduction with the WHMM. Regarding (1), the discussion regarding the demagnetization field holds significant importance in both the fields of electron holography and permanent magnets. The *H_d_* mapping derived from this method showed strong agreement with the predictions from classical electromagnetism and micromagnetic simulations. To improve the precision of phase analysis, WHMM-based noise reduction, which separates signals weaker than the noise, can be a promising solution. While narrow fringe spacing is essential for high spatial resolution, it reduces the visibility of holograms, determining the phase accuracy in electron holography. After applying the WHMM, the denoised phase image under the narrowest fringe spacing showed a significant reduction in artificial phase jumps caused by low visibility. Thus, these methods present a viable approach to overcoming technical limitations in electron holography, facilitating more precise magnetic domain analysis. Moreover, the combined techniques of phase extraction and noise reduction are expected to contribute to the further advancement of the analysis of the magnetic domain in permanent magnets.

## Figures and Tables

**Figure 1 nanomaterials-14-02046-f001:**
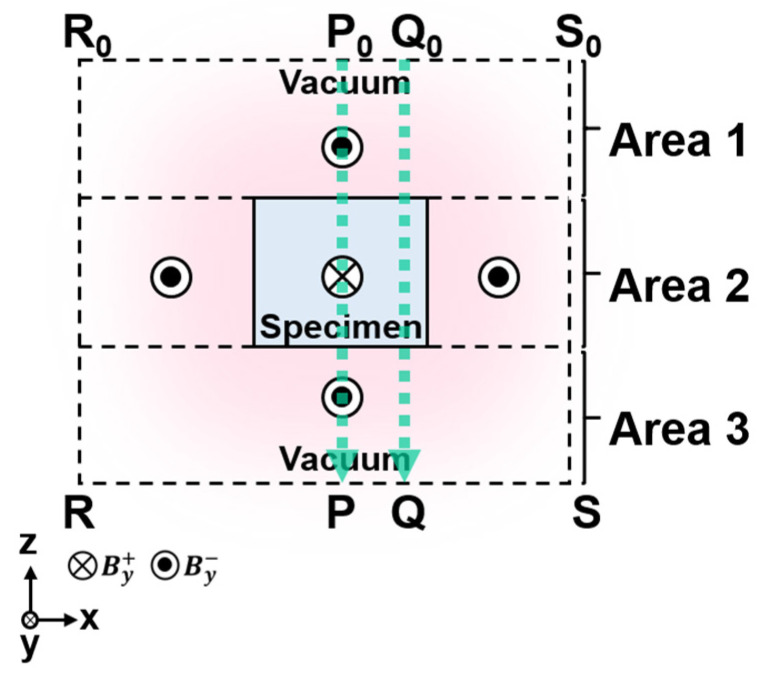
Diagrammatic cross-section of a bar magnet with magnetization aligned along the *y*-axis. Reprinted from [90], Copyright (2023) by Oxford University Press.

**Figure 2 nanomaterials-14-02046-f002:**
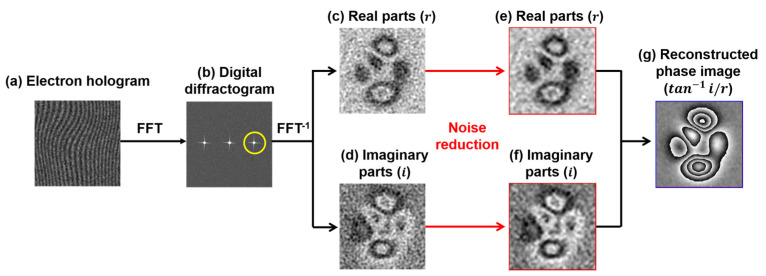
Phase retrieval process: (**a**) electron hologram; (**b**) digital diffractogram obtained through FFT of (**a**), with the yellow circle indicating a specific frequency zone selected for inverse FFT (FFT^−1^); (**c**,**d**) real and imaginary components of the complex image reconstructed from the hologram via FFT^−1^; (**e**,**f**) real and imaginary components after noise reduction applied to (c) and (d), respectively; and (**g**) reconstructed phase image derived from $\tan^{-1}i/r$. Reprinted from [89], Copyright (2024) by Springer Open.

**Figure 3 nanomaterials-14-02046-f003:**
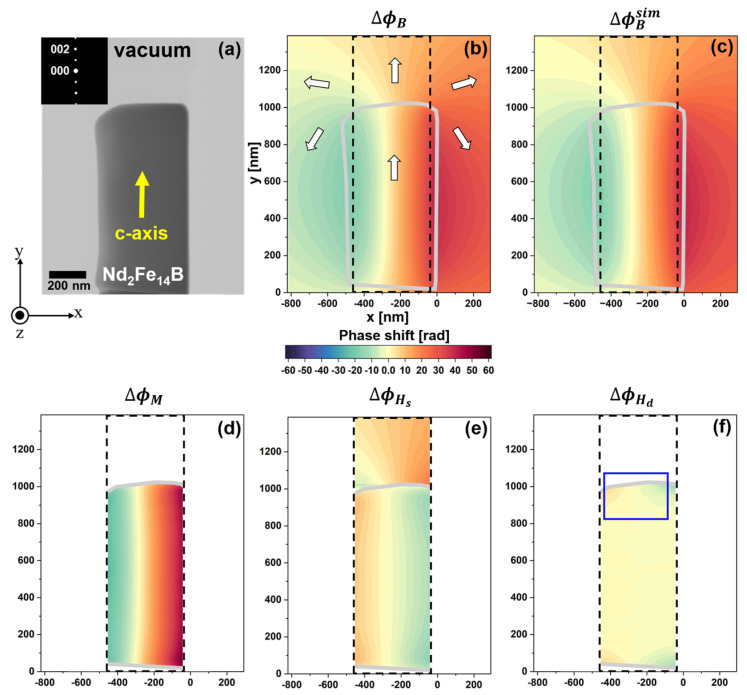
Extraction of the phase shift induced by the demagnetization field from an electron holography observation of the thin-foil Nd_2_Fe_14_B specimen: (**a**) TEM image of the thin-foil specimen; (**b**) reconstructed phase image representing ${\Delta \phi }_{B}$ via electron holography; (**c**) calculations of ${\Delta \phi }_{B}$, showing good agreement with the observation in (**b**); (**d**) phase image representing ${\Delta \phi }_{M}$; (**e**) phase image representing ${\Delta \phi }_{H_{s}}$; (**f**) phase image of ${\Delta \phi }_{H_{d}}$ within the specimen. Reprinted from [90], Copyright (2023) by Oxford University Press.

**Figure 4 nanomaterials-14-02046-f004:**
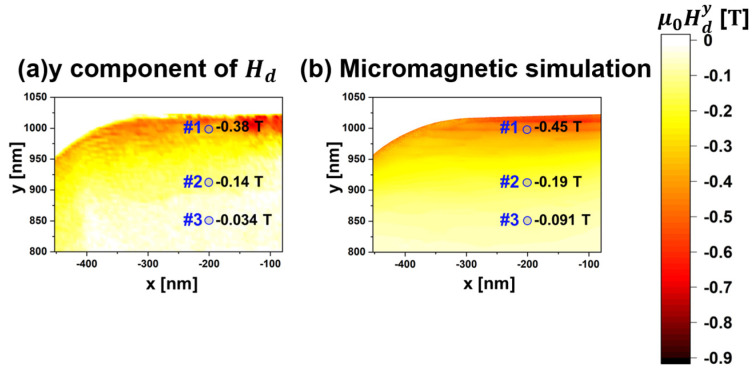
Evaluation of mapping $\mu_{0}H_{d}^{y}$ within the thin-foil Nd_2_Fe_14_B specimen: (**a**) plotting values of $\mu_{0}H_{d}^{y}$ derived from the phase images and (**b**) result of micromagnetic simulation. Reprinted from [90], Copyright (2023) by Oxford University Press.

**Figure 5 nanomaterials-14-02046-f005:**
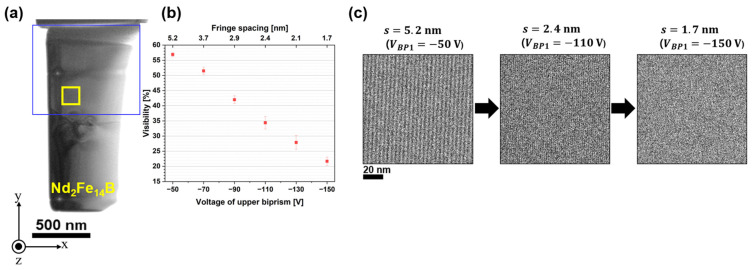
Relationship between fringe spacing and visibility of electron holograms: (**a**) TEM image of the Nd_2_Fe_14_B specimen; (**b**) visibility plotted against the upper biprism voltage (lower axis) and fringe spacing (upper axis); (**c**) series of electron holograms taken at fringe spacings of 5.2 nm, 2.4 nm, and 1.7 nm. Reprinted from [89], Copyright (2024) by Springer Open.

**Figure 6 nanomaterials-14-02046-f006:**
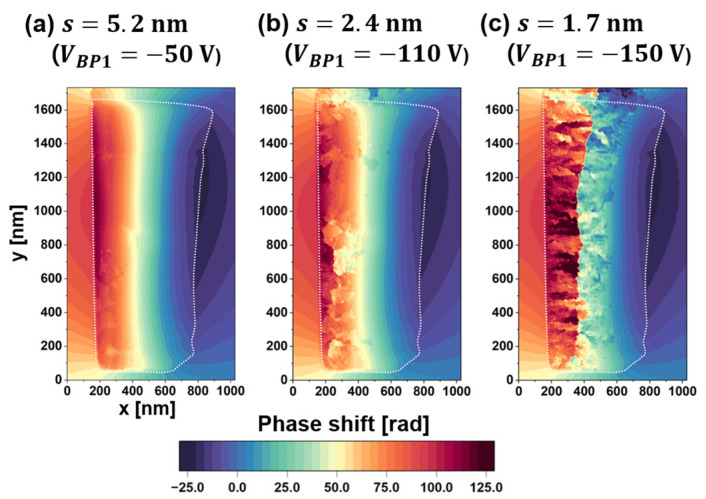
Effect of fringe spacing on the reconstructed phase image in terms of artificial phase jumps: (**a**–**c**) phase images reconstructed at fringe spacings of 5.2 nm, 2.4 nm, and 1.7 nm, respectively. Reprinted from [89], Copyright (2024) by Springer Open.

**Figure 7 nanomaterials-14-02046-f007:**
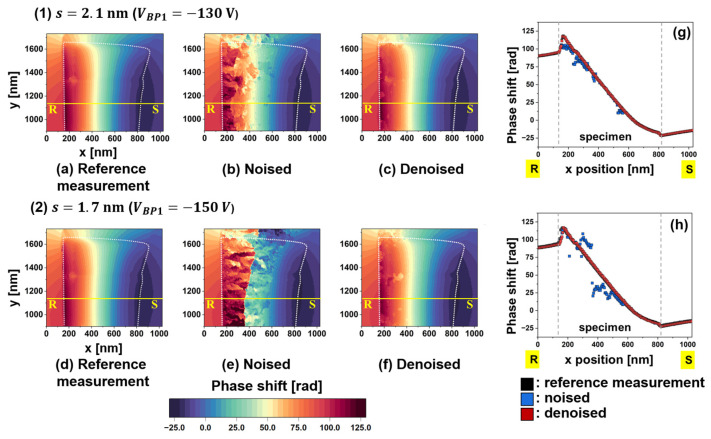
Evaluation of phase images after applying noise reduction, collected with two fringe conditions: (**1**) *s* of 2.1 nm and (**2**) *s* of 1.7 nm. (**a**,**d**) Reference-measurement image showing the negligible noise for conditions of (**1**,**2**), respectively. (**b**,**e**) Original phase images, labeled “Noised” for conditions of (**1**,**2**), respectively. (**c**,**f**) Phase images, labeled “Denoised”, after application of noise reduction for the conditions of (**1**,**2**), respectively. (**g**,**h**) Phase shift profiles measured along the R-S lines in (**a**–**c**) and (**d**–**f**), respectively. Reprinted from [89], Copyright (2024) by Springer Open.

**Figure 8 nanomaterials-14-02046-f008:**
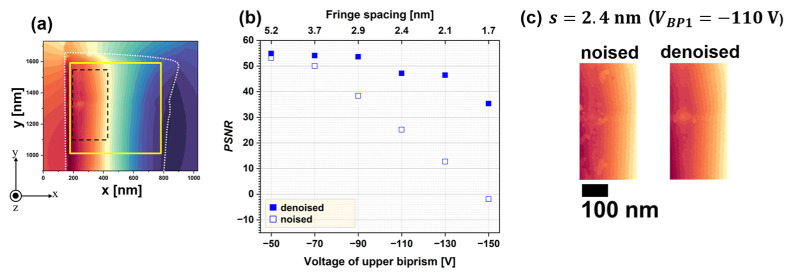
Assessment of the denoising effect via *PSNR*: (**a**) reference image at a fringe spacing of 2.1 nm; (**b**) *PSNR* as a function of the upper biprism voltage (lower axis) and fringe spacing (upper axis); (**c**) original (labeled ‘noised’, **left**) and denoised (**right**) images at a fringe spacing of 2.4 nm, enlarged from the region enclosed by the dashed lines in (**a**). Reprinted from [89], Copyright (2024) by Springer Open.

## Data Availability

No new data were created or analyzed in this study. Data sharing is not applicable to this article.

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
