# Peer review of "Electron Holography for Advanced Characterization of Permanent Magnets: Demagnetization Field Mapping and Enhanced Precision in Phase Analysis"

_nanomaterials, 2024, doi:10.3390/nano14242046_

Round 1

Reviewer 1 Report

Comments and Suggestions for Authors

This manuscript compiles the results published in Microscopy 72, 343 (2023) and Applied Microscopy 54, 4 (2024) by the same author. I do not have scientific comments as both the context and the figures have been through the peer-review processes of the other journals. I found no new information is presented in this work. 

The author should go through the text one more to fix the typos, enter space, and fix the titles in the references. 

Figure 1 has also been presented in the Microscopy paper as Fig.1a, therefore the copyright must be mentioned. 

Author Response

I sincerely appreciate the reviewer’s thorough evaluation of the manuscript and the valuable comments provided. To address the points raised by the reviewer, I have appropriately revised the manuscript. The revised parts are highlighted in red in the main text. My point-by-point responses are included below. I hope that the editor and reviewers agree with these revisions.

(1) comment-1

 The author should go through the text one more to fix the typos, enter space, and fix the titles in the references.

Reply

Thank you for your careful reading and valuable comment. I have thoroughly reviewed the manuscript and corrected the typos, adjusted spacing, which are highlighted in red in the main text as suggested.

For example,  

“Electron holography overcome~”  “Electron holography overcomes~” (page1, lines 25)

“~[first timer in Equation (1)].” “~[first term in Equation (1)]” (page13, lines 506)

(2) comment-2

 Figure 1 has also been presented in the Microscopy paper as Fig.1a, therefore the copyright must be mentioned. 

Reply

To precisely reveal the copyright of Fig.1 raised from the reviewer’s comment, I have added the following lines in Fig.1.

Reprinted from [90], Copyright (2023) by Oxford University Press.” (page6, lines 248)

Reviewer 2 Report

Comments and Suggestions for Authors

In this manuscript, a review on electron holography characterization of permanent magnets is presented. Overall, this is a well written review. Some comments to be considered during the revision are listed below. 1. As a review paper, the manuscript seemed to be a little bit short and the number of references is small. It’s better to extend the contends. 2. Could you include some contends in the aspect of application? For example, introducing why the demagnetization field is important, how can the study in demagnetization field can contribute to engineering industry. 3. There are some issues in equation format. Many equations are with extra unnecessary superscript after the integral sign, such as eq. (3), (6) (7). 4. Are there other noise suppression methods for thin foiled NdFeB images? Why the review is only focused on WHMM? It’s better to include review on other methods.

Author Response

I sincerely appreciate the reviewer’s thorough evaluation of the manuscript and the valuable comments provided. To address the points raised by the reviewer, I have appropriately revised the manuscript. The revised parts are highlighted in red in the main text. My point-by-point responses are included below. I hope that the editor and reviewers agree with these revisions.

(1) comment-1

 As a review paper, the manuscript seemed to be a little bit short and the number of references is small. It’s better to extend the contends. 

Reply

I agree with the reviewer’s comment. To address this issue, I have added the following lines briefly reviewing the current status of developing high-coercivity of Nd-Fe-B sintered magnet, which is a significant focus of this review article.

For example, the coercivity of sintered magnet is approximately 0.2 T at the operating temperature of traction motors (~473 K). Enhancing  can be achieved by substituting Dy for Nd, resulting in the coercivity of up to 3 T at room temperature due to the improved magnetocrystalline anisotropy. However, this substitution decreases the saturation magnetization of the magnet due to the antiferromagnetic spin coupling between Dy and Fe [35-38]. Furthermore, from an industrial perspective, Dy is classified as a critical element with limited natural availability, raising concerns about its sustainable use [39,40].” (page2, lines 69)

In addition, this article reviewed the study on demagnetization field that is strongly related to the coercivity. Therefore, I have added the details about the coercivity mechanism and advantages of electron holography to help understanding on coercivity mechanism, as follows.

In principle, coercivity represents the critical magnetic field required to induce undesired magnetization reversal. For magnetic domains to reverse under external magnetic fields or thermal fluctuations, the energy barrier associated with Ku must be overcome. However, magnetization reversal is a complex phenomenon, as it depends not only on the crystallographic microstructure but also on magnetic domain structures, which are sensitive to magnetic dipolar interactions and exchange interactions between neighboring domains. This complexity poses significant challenges for analyzing the coercivity mechanism [41]. The coercivity is determined by the path of least resistance in these mechanisms, in which the magnetization reversal can occur either continuously through a coherent or incoherent rotation, or discontinuously through a dynamic domain motion. Achieving high coercivity requires impeding magnetization rotation through strong magnetic anisotropy and inhibiting the nucleation or the growth of reverse magnetic domains [42]. Especially, the  of Nd-Fe-B sintered magnets is predominantly governed by the nucleation mechanism. Hence, reverse magnetic domains tend to nucleate preferentially in the regions of locally weak magnetocrystalline anisotropy, such as the surface of the Nd2Fe14B grains[43,44]. Therefore, Electron holography enables high-resolution analysis of magnetic domain structures by providing detailed information about the magnetization distribution and the direction and strength of the magnetic flux density (B) at the nanometer scale, thereby contributing to a deeper understanding on coercivity mechanism. In addition to electron holography, Lorentz microscopy is another powerful TEM-based technique for imaging magnetic domain structures. By exploiting the lateral deflection of incident electrons due to the Lorentz force, Lorentz microscopy (in Fresnel mode) visualizes magnetic domain walls as black and white contrast lines, resulting from the electron deficiency and excess in defocused conditions [45,46]. This capability allows for in-situ observation of domain wall motion during magnetization reversal. However, Lorentz microscopy faces challenges in providing quantitative information of magnetic field and high-contrast imaging of domain wall under complex magnetization distributions, particularly those with varying magnetic flux directions across domain walls. Other techniques, including micro-SQUID magnetometry [47], magnetic force microscopy (MFM) [48,49], photoemission electron microscopy (PEEM) [50], are also employed to investigate local magnetic properties. However, these methods often lack the spatial resolution needed to observe magnetic domain structures in nanometer-sized materials or to resolve intricate features, such as vortex-core structures.” (page2, lines 84)

To further emphasize and remind the importance of study on demagnetization field mapping using electron holography, the following lines have been added.

As elucidated in the introduction, electron holography allows for determination of phase shifts induced by B [1,2]. However, isolating and analyzing Hd, which represents only one component of B, remains challenging. This technical constraint (i.e., the inability to directly observe Hd ) is also problem for other tools such as XMCD [65,66], SP-STM [67], micro-SQUID magnetometry [68], magnetic force microscopy [69], Lorentz microscopy, and differential phase contrast microscopy [70-72]. Moreover, understanding Hd is crucial for developing high coercivity in permanent magnets. The Hd significantly influences the coercivity mechanism, as regions with a high distribution of Hd within the magnet can initiate the nucleation of reverse magnetic domains, resulting in a reduction in coercivity [53,54]. The role of Hd in the coercivity mechanism has been explored through micromagnetic simulations, primarily focusing on simple model specimens [54,57-62]. However, investigations of Hd in real magnetic specimens remain limited due to the technical constraints of experimental tools.

This section introduces a method for mapping Hd using phase images, providing a potential solution to address challenges associated with Hd in both the field of electron microscopy and magnetic material engineering.” (page 7, lines 276)

Therefore, incorporating the aforementioned content, I have extended the length of the manuscript and increased the number of references while addressing the comments including comment-1. The added and revised references are highlighted in red in Reference section of the main text.

 (2) comment-2

 Could you include some contends in the aspect of application? For example, introducing why the demagnetization field is important, how can the study in demagnetization field can contribute to engineering industry.

Reply

To address the reviewer’s comment, I have revised and added the following lines to emphasize contribution and importance of demagnetization from the perspective of developing high coercivity of permanent magnet.

One of critical aspect of the coercivity mechanism is the distribution of Hd within Nd-Fe-B systems, as Hd contributes to undesired magnetization reversal [51-54]. While the magnitude and distribution of Hd in sintered magnets are influenced by factors such as the shape, size, and orientational dispersion of the Nd2Fe14B crystal grains, an increase in Hd facilitates the nucleation of reverse magnetic domains, ultimately degrading the coercivity [53,54]. The influence of Hd on the magnetization reversal has been investigated through micromagnetic simulations, based on Landau-Lifshitz-Gilbert calculations [55,56], which primarily focused on model specimen with simplified geometries [54,57-62]. Li et al. [54] showed that the edges and corners of crystal grains, modeled as polyhedral, serve as potential nucleation sites for magnetization reversal. This phenomenon is closely associated with the distribution of the demagnetization field, which is particularly pronounced on the c-plane surfaces of the grains. Bence et al. [58] investigated the demagnetization field by calculating its distribution in artificial crystal grains with diameters ranging from 55 nm to 8.3m, focusing on the influence of surface grains on magnetization reversal. Despite these simulation studies, experimental methods capable of directly analyzing Hd in real magnets remain insufficient. Understanding the distribution of Hd within magnets provides a crucial indicator for effectively enhancing their coercivity. For example, as mentioned earlier, reverse magnetic domains preferentially nucleate in regions with locally reduced magnetocrystalline anisotropy, such as the surface of Nd2Fe14B grains. To improve this problem, substituting Nd sites in the Nd2Fe14B lattice with heavy rare earth (HRE) elements, such as Dy and Tb, only on the surfaces of the crystal grains can locally increase the magnetic anisotropy field at nucleation sites by forming HRE-rich shell with high anisotropy field [43,44,63]; this method is known as heavy rare earth grain boundary diffusion process (HRE-GBDP) [64]. This effect of magnetically “hard” HRE-rich shell can maximize when we know where the Hd is strongly distributed within the magnet. Following the discussion by Li et al. [54], introduction of the HRE-rich shell only at c-plane surface of the grain, showing the strong Hd distribution, required the significant external field for nucleation of reverse magnetic domains. Hence, the discussion about Hd is important for developing high coercivity of Nd-Fe-B magnet.

Discussion about Hd is also essential for electron holography study as well as for the industry of permanent magnet.” (page3, lines 117)

To avoid redundancy, the following lines have been removed from the main text.

A key aspect of the  mechanism is the internal distribution of the demagnetizing field () within the magnet, as  can drive undesirable magnetization reversals [38-41]. Li et al. [41] observed that crystal grain edges and corners, modeled as polyhedra, can be potential nucleation sites of reverse magnetic domains, largely due to the high demagnetization field on the c-plane surfaces. Similarly, Bance et al. [42] investigated artificial grains of varying dimensions (from 55 nm to 8.3 μm), demonstrating that surface grains can influence magnetization reversal. Analyzing magnetic domain behavior is therefore crucial for advancing the performance of Nd-Fe-B magnets.” (page2, lines 75)

 (3) comment-3

 There are some issues in equation format. Many equations are with extra unnecessary superscript after the integral sign, such as eq. (3), (6) (7). 

Reply

As noted in you comment, unnecessary superscripts are preset in Eq.(3),(6) an (7). However, as these superscripts are blank, they will not appear in the final published version of the paper. Unfortunately, the integral equation format in Word does not provide the option to remove superscripts alone. Nevertheless, given the importance of this issue raised from the reviewer’s comment, I will carefully review the equations in future process to ensure that they are printed in the intended presentation.

 (4) comment-4

 Are there other noise suppression methods for thin foiled NdFeB images? Why the review is only focused on WHMM? It’s better to include review on other methods.

Reply

I agree with the reviewer’s comment on the need for a more comprehensive review of other noise reduction methods. However, there are only a limited number of reports specifically adressing the application of noise reduction techniques to Nd-Fe-B magnets. As discussed in the main text, this article focuses on noise reduction as a strategy to enhance the precision of phase analysis, which can be useful for analyzing magnetic domain structure of magnetic materials. Consequently, Nd-Fe-B magnets can be a good target to effectively demonstrate the impact of noise reduction, as shown in the main text. Therefore, the following lines have been revised and expanded to include a review of other denoising methods and WHMM applied to holograms. These revisions also aim to explain the potential of these methods as proming tool for improving phase analysis in the obsrvation of magnetic domains.

Applying noise reduction techniques through image processing can greatly enhance the accuracy of phase analysis without altering optical or interferometry parameters. In principle, the long exposure time for achieving higher electron dose may be the easiest approach to improve the phase detection limit. However, this condition can lead to unwanted events, such as surface contamination and specimen drift. Therefore, imaging processing, including sparse coding [75-77] and tensor decomposition [78], have proven effective for low-dose holograms and in-situ experiments. Tensor decomposition reduces noise by separating the data into low-rank components and residual noise. The low-rank components capture the dominant structural or physical features of the data, while high-frequency noise is isolated in sparse or less significant components, the noise reduction using tensor decomposition can be effectively performed, preserving the essential information[79-81]. As a result, Nomura et al. [78] reported that employment of tensor decomposition successfully extracted the reasonable phase information from low-dose electron hologram of p-n junction. Also, sparse coding enhances electron holograms by representing the data through a sparse combination of significant patterns (learned dictionary elements) [82]. This method emphasizes essential signals while reducing irrelevant or noisy components [82-84]. Following the study by Anada et al. [75,76], the electrostatic potentials in GaAs-based p-n junctions was clearly visualized from the low-dose holograms. Takahashi et al. [77] showed that the denoising using sparse coding successfully removed the inevitable phase jumps in phase image of Pt nanoparticle, which make it much more difficult to quantitatively analyze local phase values and details about this will be discussed in Section 4.” (page4, lines 169)

Tamaoka et al. [88] utilized WHMM in the analysis of experimentally obtained holograms from a multilayered LaFeO3/SrTiO3 film. Noise reduction improved the quality of the reconstructed phase image, representing a gap in the electrostatic potential at the non-magnetic LaFeO3/SrTiO3 interface. Despite most previous studies on noise suppression focusing primarily on non-magnetic materials, these denoising methods including WHMM holds significant potential for broader application in electron holography, particularly in the observation of magnetic domain structures. Note that the Nd2Fe14B crystals present challenges for phase analysis related to magnetic domains due to poor contrast image caused by heavy Nd element absorption. Therefore, noise reduction can be promising method to improve the magnetic domain analysis for the Nd-Fe-B based magnets.” (page4, lines 203)

To avoid redundancy, the following lines have been removed from the main text.

For example, Anada et al.[53,54] demonstrated the utility of sparse coding to visualize electrostatic potentials in GaAs-based p-n junctions. Nomura et al.[55] utilized tensor decomposition to extract phase information effectively, even in low-electron-dose scenarios.” (page4, lines 190)

However, this article primarily focuses on the WHMM method. Conventioanl WHMM and other previously reported mehtods has typically been applied to electron holograms. In contrast, the WHMM mainly reviwed in this article was specifically utilized for complex images, solving a criticial tehcnial dilemma in electron holography (i.e., relationship between spatial coherence and visibility). To further emphasize this distinct approach, I have revised and added the following lines.

“To enhance the quality of the phase image, a technique of noise reduction can significantly improve the precision of the method, enabling a more in-depth analysis of Hd. To date, numerous studies have demonstrated improved precision in phase analysis by applying sparse coding [75-77], tensor decomposition [78], and the WHMM [85,88,89] to electron holograms. Notably, as explained in Section 1, there is a dilemma regarding interferometric parameter: while a narrow s is essential for achieving high spatial resolution in phase images, it simultaneously leads to a deterioration of phase resolution due to reduced Vobs [1]. To overcome this technical limitation, denoising using WHMM was applied to the complex images of Nd2Fe14B thin-foil, with variations in the the fringe spacing of holograms, distinguishing it from other denoising method typically used for electron holograms. Therefore, this session reviewed the effectiveness of WHMM denoising to complex images of a thin-foiled Nd-Fe-B magnet, highlighting its capability to suppress the artificial phase jumps induced from insufficient Vobs.” (page10, lines 414)

Round 2

Reviewer 1 Report

Comments and Suggestions for Authors

The manuscript can be accepted for publication. 

Reviewer 2 Report

Comments and Suggestions for Authors

My comments have been well replied and considered.